# Investigating the Impact of Displacement Cascades on Tritium Diffusion in MgT_2_: A Molecular Dynamics Study

**DOI:** 10.3390/ma16093359

**Published:** 2023-04-25

**Authors:** Siwei Zhang, Size Chen, Dan Xiao, Chao Wang, Haixia Wang, Yong Zhang, Taosheng Li

**Affiliations:** 1Hefei Institutes of Physical Science, Chinese Academy of Sciences, Hefei 230031, China; siwei.zhang@inest.cas.cn (S.Z.); size.chen@inest.cas.cn (S.C.);; 2University of Science and Technology of China, Hefei 230026, China; 3National Synchrotron Radiation Laboratory, University of Science and Technology of China, Hefei 230026, China

**Keywords:** molecular dynamics, MgT_2_ target, displacement cascade, diffusion

## Abstract

Molecular dynamics methods were utilized to investigate displacement cascades and tritium diffusion in α-MgT_2_. It was observed from collision cascades results that the stable number of defects weakly depended on temperature, while the peak and stable number of defects linearly increased with increasing the primary knock-on atom energy. The results of the mean square displacement study revealed that defects had a significant impact on tritium diffusion. The clustering of magnesium self-interstitial atoms and diffusing tritium atoms results in an increased diffusion barrier, whereas the formation of clusters between tritium interstitial atoms is relatively difficult and has no significant impact on the diffusion barrier. The presence of magnesium and tritium vacancies has a minimal effect on the diffusion barrier due to the large number of diffusing tritium atoms that offset the adsorption of vacancies on diffusing atoms. Both magnesium and tritium interstitial atoms increase the collision probability of diffusing atoms, leading to an increased diffusion prefactor. Magnesium vacancies cause significant lattice distortion, increasing the diffusion barrier, while the impact of tritium vacancies on the diffusion barrier is small due to their minimal lattice distortion effect. The research uncovered significant disparities in the diffusion properties of hydrogen and tritium, indicating that the results of the study of hydrogen storage could not be applied to tritium.

## 1. Introduction

The deuterium (D)–tritium (T) neutron source has numerous applications, including advanced manufacturing, contraband detection, advanced medicine, and oil well logging [1,2]. However, the tritium target represents a bottleneck in enhancing neutron source intensity, primarily due to limited deuterium ion incident range and thermal stability [3]. Magnesium hydride (MgH_2_) is a promising material for tritium targets, owing to its good hydrogen storage capacity and stable thermodynamic performance [4]. Tritium targets in neutron sources are subjected to intense deuterium ion radiation, which can cause the formation of point defects or defect clusters in the material. These defects may further evolve into microstructural flaws, ultimately affecting the service lifetime of the tritium target by influencing the extent of tritium release.

Magnesium has been known to be highly susceptible to irradiation damage [5]. Both simulation and experimental studies have shown that when subjected to different ion irradiation, including electrons, Kr^3+^, and C^6+^, magnesium may develop both microscopic damages, such as vacancies, interstitial atoms, and dislocations, as well as macroscopic wave structures [5,6,7,8].

Studies have demonstrated that point defects such as vacancies and self-interstitial atoms can trap hydrogen atoms, reducing the diffusivity [9]. The diffusion of hydrogen in a perfect MgH_2_ lattice has been the subject of previous studies. Ramzan calculated the mean square displacement (MSD) of deuterium in bulk MgH_2_ and determined that the deuterium diffusion constant in bulk MgH_2_ is 2.33 × 10^−9^ m^2^/s [10]. Yao also calculated the diffusivity of hydrogen in MgH_2_ at different temperatures, with results indicating that the diffusivity of hydrogen in MgH_2_ ranges from 10^−18^ to 10^−24^ m^2^/s at temperatures between 300 and 100 °C [11]. However, the impact of different types of irradiation damage and densities of irradiation damage on hydrogen diffusion has yet to be studied. As a result, it is crucial to study the diffusion of hydrogen in MgH_2_ containing crystal damage.

Given that the evolution of primary irradiation damage occurs on the picosecond time scale, molecular dynamics (MD) simulations are appropriate for obtaining microscopic insights. The present study employs MD simulations to investigate radiation damage events in bulk MgT_2_. In this work, we focused on the formation of vacancies and interstitial atoms in MgT_2_ under irradiation through MD simulations. This is followed by a presentation of the results on the impact of radiation damage on tritium diffusion, which were determined using the MSD method.

## 2. Materials and Methods

The classical molecular dynamics simulations were carried out using the Large-scale Atomic/Molecular Massively Parallel Simulator (LAMMPS), accompanied with the angular dependent potential (ADP) empirical potential [12]. This empirical potential, which is a generalization of the Embedded Atom Method (EAM) potential, was chosen for atomic interactions because it accurately describes the stable structures of some existing magnesium hydrides, such as α-MgH_2_ (P4_2_/mnm) and γ-MgH_2_ (Pbcn), and it allows for the simulation of hydrogen behavior in magnesium, including the energies of different point defects and stacking faults in magnesium. Additionally, the MSD of hydrogen in magnesium was assessed using the aforementioned potential. The accuracy of the MSD description was found to be on the same order of magnitude.

The classical molecular dynamics simulations of the displacement cascade were performed using simulation boxes with a bcc structure α-MgT_2_ containing 750,000 atoms, 250,000 magnesium atoms, and 500,000 tritium atoms, with dimensions of 50 a_0_ × 50 b_0_ × 50 c_0_. The lattice constant values for a_0_, b_0_, and c_0_ are 4.5119 Å, 4.5119 Å, and 3.0157 Å, respectively, yielding a cuboid simulation box with side lengths of 225.59 Å, 225.59 Å, and 150.79 Å. The simulations were carried out using PKA recoil energies of 1, 2, and 5 keV.

Prior to initiating the cascade, the system was equilibrated for 30 ps at 300 K using the canonical ensemble (NVT). The cascade was then initiated by imparting kinetic energy to the selected primary knock-on atom (PKA), which was located in the center of the system. At least 6 molecular dynamics (MD) runs were performed for each irradiation case.

The leapfrog algorithm was utilized to resolve the equations of motion of atoms. To avoid the issue of atoms being displaced too far, a variable time-step method was employed with a displacement limit of 0.005 crystal lattice per time-step. The simulations were performed under the microcanonical ensemble (NVE) with periodic boundary conditions. The results were visualized and processed using the OVITO [13] software package. The Wigner–Seitz defect analysis was utilized to characterize interstitials and vacancies, and the initial system prior to equilibration was used as a reference for the analysis [14,15]. The coordination analysis was used to compute the radial distribution function (RDF).

In order to examine the impact of defects on the diffusion of hydrogen in the MgT_2_, a systematic investigation was carried out to evaluate the diffusion coefficient of hydrogen atoms under varying temperature conditions. This was achieved by analyzing the MSD of hydrogen atoms, which was calculated using the following equation:(1)$$\mathrm{MSD}=\frac{1}{n}\sum_{i=1}^{n}{\left[r_{i}(t)-r_{i}0\right]}^{2}$$
where *t* is the diffusion time, *r* is the vector position of the tritium atom, *i* is the ID of the atoms, and *n* is the number of atoms. The Einstein equation from random walk theory was used to calculate the diffusivity of tritium:(2)$$D=\frac{1}{6}\underset{t\to \infty }{\lim}\frac{d}{dt}(\mathrm{MSD})$$

Initially, we modeled the diffusivity of tritium in a perfect lattice using a 10 a_0_ × 10 b_0_ × 10 c_0_ supercell containing 2000 magnesium atoms and 4000 tritium atoms. A total of 20 tritium atoms were randomly generated in the system and simulated for a period of 50 ps at temperatures ranging from 300 to 750 K. To increase accuracy, the simulation was repeated four times, and the diffusivity was determined based on the Einstein equation from the random walk theory. The diffusivity and temperature can be represented by the Arrhenius equation:(3)$$D=D_{0}exp(-\frac{E_{a}}{\kappa_{B}T})$$
where *D_0_* is the diffusion prefactor, *Ea* is the diffusion energy barrier, *κ_B_* is the Boltzmann constant, and *T* is the temperature. Taking the natural logarithm of each side of the equation is performed as follows:(4)$$ln(D)=lnD_{0}-\frac{E_{a}}{\kappa_{B}}\frac{1}{T}$$

It was observed that *ln*(*D*) exhibits linear behavior with respect to *1*/*T*. The diffusivity of tritium atoms at various temperatures can be obtained from the MSD results, and the diffusion prefactor and diffusion energy barrier can then be determined by a linear fit of *ln*(*D*) to *1*/*T*.

Furthermore, the diffusion of tritium in supercells with varying concentrations of self-interstitial atoms (SIAs) and vacancies was simulated. By randomly introducing or removing tritium or magnesium atoms in a system, various types and densities of defects can be generated. Subsequent relaxation of the system causes the added atoms to diffuse and occupy stable positions, creating interstitials. Meanwhile, the neighboring atoms of the removed atoms move into the vacant positions, leading to the formation of vacancies. This process allows for the creation of systems with different defect densities. As the initial system is a perfect lattice, the added atoms will not be absorbed by the system, thus ensuring that the density of interstitials remains constant during the simulation. Similarly, the vacancies will not be filled by normal atoms due to lattice distortions after relaxation, thereby ensuring that the concentration of defects remains unchanged throughout the simulation. Then, a total of 20 tritium diffusion atoms were randomly introduced into the system. All other simulation conditions, including the simulation time and temperatures, remained unaltered from those of the simulation in the perfect supercell. By comparing the diffusivity, diffusion prefactor, and diffusion energy barrier of tritium in both the defective and perfect supercells, the study aimed to determine the impact of defects on tritium diffusion.

## 3. Results

### 3.1. Production of Point Defects after Displacement Cascade

The average number of Frenkel pairs during displacement cascades at three temperatures and three energies are presented in Figure 1. The results reveal that the cascades under different conditions exhibit similar features: At the onset of the cascade, a significant increase in the number of Frenkel pairs is observed. This increase reaches its maximum during the thermal spike phase, which occurs at 0.3 ps in Figure 1a and 0.4 ps in Figure 1b. Subsequently, during the annealing phase the vacancies and SIAs in Frenkel pairs undergo recombination, resulting in a gradual reduction in the number of defects. Finally, the system attains a stable state with a substantially reduced number of defects.

Based on Figure 1a, a distinct trend in the formation of defects with increasing temperature is observed. Specifically, Figure 1a illustrates a gradual rise in the peak number of defects as the temperature increases. Notably, the peak number at 400 K shows only a marginal increase relative to that at 300 K, whereas the peak number at 500 K exhibits a significant increase. This is attributed to the fact that at 500 K, the temperature is close to the decomposition temperature of MgT_2_, which results in intensified hydrogen atom motion, resulting in more atoms being displaced, leading to a significant rise in the peak number of defects. However, the discrepancy in the number of stable defects is not substantial, and the quantity of the stable number of defects at 300 K is slightly greater than that at 400 K and 500 K. This phenomenon can be explained by the intensified motion of atoms at higher temperatures, which leads to a higher probability of vacancies and SIA recombination.

Figure 2a–c depict the projection views of the thermal spike phase at different temperatures. Despite the increase in the number of defects as the temperature rises, the projection area does not show significant growth. This is because a high temperature increases the probability of vacancies and SIA recombination, resulting in the increased number of displaced atoms remaining close to the vicinity of the PKA atom’s trajectory. This leads to a greater number of defects on the PKA trajectory, but the final projection area does not experience a significant increase, as evidenced by Figure 2d,e. Notably, Figure 2 shows that tritium atoms generate more Frenkel pairs than magnesium atoms, likely due to tritium atoms having a lower threshold displacement energy and thus being more effective at inducing defects.

Figure 1b depicts the progression of Frenkel pairs in MgT_2_ over time at different energies. Figure 1b indicates that as the PKA energy increases, the damage level also increases and the time required to reach the peak number of defects increases accordingly. This is due to the fact that higher PKA energy results in more cascade particles reaching the threshold displacement energy and thus generating a greater number of defects. The increase in PKA energy can also be seen to cause a rise in cascade volume, leading to a longer time required to reach both the peak number and the stable number, as shown in Figure 3. By examining the enlarged depiction in Figure 1b, it is apparent that the quantity of defects at the 40 ps–70 ps position remains relatively constant, implying that the system has stabilized.

The number of Frenkel pairs produced by each cascade damage process can be expressed by the following NRT formula [16]:(5)$$N_{NRT}=0.8E_{p}/2{\bar{E}}_{d}$$
where *E_d_* is the average threshold energy of Frenkel pair formation. Based on Equation (5), it can be observed that the average displacement threshold energy (*E_d_*) of atoms remains constant for a given system. Consequently, the quantity of defects is directly proportional to PKA energy. The relationship between the peak and stable number of Frenkel pairs and PKA energy is presented in Figure 4. The results show a linear correlation between the two variables. The slope of the curve representing the peak defect number is 4.0 times steeper than the slope of the stable number of defects.

The formation of Frenkel pairs due to irradiation has a significant impact on tritium diffusion and ultimately leads to the release of tritium in macroscopic experiments. To evaluate the effect of defects on tritium release, we analyzed the variation in tritium density as a function of distance from magnesium atoms using radial distribution function (RDF). The calculation allowed for the determination of tritium release characteristics for different levels of irradiation damage. The radial distribution function is mathematically represented as follows:(6)$$g(r)=\rho (r)/\rho_{a}$$

This paper calculates the variation in tritium density as a function of distance from magnesium atoms for PKA energies of 1 keV, 2 keV, and 5 keV, and the results are depicted in Figure 5.

According to the results shown in Figure 5, the radial distribution function (RDF) of hydrogen atoms during displacement cascade processes exhibits a decreasing trend. This decreasing trend increases with increasing PKA energy. This implies a decrease in the density of hydrogen atoms surrounding magnesium atoms, which is mainly attributed to the overall movement of hydrogen atoms in the irradiated region towards the system boundary. Since the displaced tritium atoms only account for a small portion of the total number (500,000) of tritium atoms, the decrease in RDF is not significant. However, it can be inferred that in the experimental process, more irradiation by high-energy particles and an increase in defects within the system will lead to more severe migration of tritium within the system.

### 3.2. Tritium Diffusion in α-MgT_2_

#### 3.2.1. Tritium/Hydrogen Diffusion in Perfect α-MgT_2_/α-MgH_2_

In order to determine the diffusivity of tritium in perfect α-MgT_2_, the MSD of tritium was studied at temperatures ranging from 300 K to 750 K. Additionally, to examine the impact of the isotope effect on diffusion, the diffusion coefficient of hydrogen (tritium is the isotope of hydrogen) in MgH_2_ was calculated and the results are presented in Figure 6.

As demonstrated in Figure 6a,b, the MSD results of tritium and hydrogen exhibit linear behavior with time, which is in agreement with the theoretical analysis. The diffusivity was then calculated based on the MSD results and is presented in Table 1. It can be observed that the diffusivity of tritium/hydrogen in perfect α-MgT_2_/α-MgH_2_ increases as the temperature increases, with the diffusivity of hydrogen being significantly higher than that of tritium.

To determine the diffusion energy barrier and prefactor for tritium/hydrogen, the natural logarithm of diffusivity (*lnD*) was studied as a function of the inverse of temperature (*1/T*), as shown in Figure 6c. It can be observed that the *lnD* is linearly related to temperature below 600 K. In the range of 650–700 K, the increase in the *lnD* becomes more significant with increasing temperature. This is due to the decomposition of MgT_2_ above 600 K, resulting in an increase in tritium diffusivity.

A linear fit was performed for the 300–600 K range, and the results are displayed in Figure 6d. Using Equation (4), we obtained a diffusion prefactor of 4.11 × 10^−10^ m^2^/s and a diffusion energy barrier of 0.058 eV for hydrogen, and a diffusion prefactor of 2.66 × 10^−10^ m^2^/s and a diffusion energy barrier of 0.056 eV for tritium.

It can be observed that the diffusion energy barriers of hydrogen and tritium are relatively comparable; however, the diffusion prefactor for tritium is approximately 64% of that of hydrogen. This implies that the diffusion of tritium is significantly smaller compared to that of hydrogen. Consequently, the findings of the research on hydrogen storage in magnesium cannot be directly extrapolated to tritium.

#### 3.2.2. Tritium Diffusion in α-MgT_2_ with SIA and Vacancies

Figure 7a,b present the Arrhenius plots of tritium diffusion in MgT_2_ at different SIA concentrations for different tritium interstitial and magnesium interstitial systems. As shown in Figure 7a, the diffusivity decreases with magnesium interstitial atoms and increases with tritium interstitial atoms. This is because the presence of magnesium interstitial atoms can trap hydrogen atoms, thus reducing the diffusivity, while tritium interstitial atoms occupy the original hydrogen diffusion site, leading to an increase in the tritium diffusivity.

Additionally, from Figure 7a,b, it can be observed that SIAs have a substantial impact on the diffusivity of tritium in MgT_2_ at low temperatures. However, this effect decreases with increasing temperature, and when the temperature is above 600 K, the SIAs have a negligible effect on the diffusivity. This can be explained as follows: at low temperatures, tritium atoms trapped by SIAs will escape as the temperature rises.

The results of the diffusivity within the temperature range of 300 K to 600 K were selected for linear regression analysis. The analysis is depicted in Figure 7c,d. The diffusion prefactor and the diffusion energy barrier were derived from the intercept and slope of the linear function and are presented in Table 2. The results in Table 2 reveal that the inclusion of interstitial atoms enhances the pre-exponential factor. Specifically, the presence of magnesium interstitials raises the prefactor from 2.66 × 10^−10^ m^2^/s to 3.65 × 10^−10^ m^2^/s, while tritium interstitials elevate it from 2.66 × 10^−10^ m^2^/s to 3.08 × 10^−10^ m^2^/s. This phenomenon can be attributed to the chemical reaction collision theory, which states that interstitial atoms augment the probability of collisions between diffusing atoms and other atoms in the system, resulting in higher diffusion prefactors. Furthermore, the data in Table 2 illustrate that magnesium interstitials heighten the diffusion barrier from 0.056 eV for a perfect lattice to 0.071 eV, whereas the presence of hydrogen interstitials has no impact on the diffusion barrier. Magnesium interstitials adsorb tritium diffusion atoms, whereas tritium interstitials do not adsorb to diffusion atoms that are also tritium. This is because magnesium interstitials have a higher binding energy with tritium atoms compared to tritium interstitials.

MSD of tritium in MgT_2_ was conducted for varying vacancy concentrations. The outcomes are depicted in Figure 8. It can be seen that tritium diffusivity decreases at different vacancy concentrations. The results of the diffusivity within the range of 300 K to 600 K were selected for linear regression analysis. Based on our calculations, we have determined the diffusion prefactor and energy barrier, which are presented in Table 3.

The results suggest that the effect of magnesium and tritium atomic vacancies on the energy barrier can be disregarded. Although vacancies can form vacancy–tritium clusters and increase the energy barrier due to the strong binding energy between tritium atoms and vacancies, this effect weakens with the increasing number of diffusing tritium atoms. In the current simulation, with 20 diffusing tritium atoms, the inhibitory effect of vacancies on tritium diffusion can be neglected, which also accounts for the insignificant influence of vacancies on the diffusion barrier.

Interestingly, we found that hydrogen vacancies significantly reduce the diffusion prefactor, while magnesium vacancies have a minimal impact. This observation aligns with the chemical reaction collision theory. Hydrogen vacancies only slightly distort the lattice due to the small size of tritium atoms, resulting in minimal changes to the atomic distribution within the system. This reduction in the number of atoms in the system decreases the collision probability of diffusing atoms, thereby leading to a decrease in the diffusion prefactor. Conversely, atoms surrounding magnesium vacancies are tritium atoms, and the lattice distortion effect is more pronounced due to the larger size of magnesium atoms. This effect leads to a higher density of atoms within the system. Despite a decrease in the number of atoms in the system, the collision probability of diffusing tritium atoms actually increases due to the rising atomic density, resulting in a slight increase in the diffusion prefactor.

## 4. Conclusions

This study employed molecular dynamics simulations to investigate the impact of temperature and PKA energy on radiation defects during cascade collisions, as well as the effects of defects on tritium diffusion in MgT_2_. The results revealed that increasing the temperature led to more severe atomic motion, which resulted in a higher peak number of defects. However, temperature also promoted defects recombination, which had a relatively small impact on the stable defect concentration. The peak and stable number of defects both exhibited linear growth within the range of 1–5 KeV PKA energy. Moreover, the study found that different types of defects had different effects on tritium diffusion. The formation of clusters between diffusing atoms and magnesium SIAs increased the energy barrier, whereas clusters between diffusing atoms and tritium SIAs had no significant impact. The formation of clusters between vacancies and tritium atoms reduced the diffusivity. Nevertheless, the high concentration of diffusing tritium atoms impeded the vacancies’ ability to adsorb diffusing atoms, leading to a negligible alteration of the diffusion barrier. Chemical reaction collision theory was used to explain the changes in the diffusion prefactor. The presence of magnesium and tritium SIAs increased the collision probability of diffusing atoms, resulting in an increase in the diffusion prefactor. The presence of magnesium vacancies caused notable distortion in the lattice structure, resulting in an increased atomic density which led to a slight elevation in the diffusion prefactor. On the other hand, tritium vacancies had a minimal effect on lattice distortion, thus reducing the collision probability and, subsequently, the diffusion prefactor. Furthermore, the study revealed significant differences in the diffusion parameters between hydrogen and tritium. Therefore, the results of the magnesium–hydrogen study cannot be applied to tritium.

## Figures and Tables

**Figure 1 materials-16-03359-f001:**
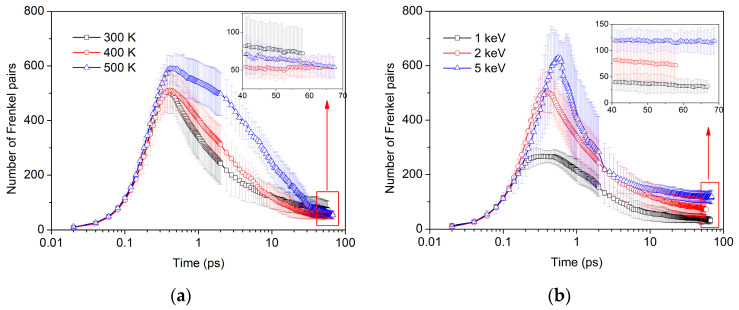
Time evolution of Frenkel pairs number produced at different temperatures and energies: (**a**) different temperatures and (**b**) different energies.

**Figure 2 materials-16-03359-f002:**
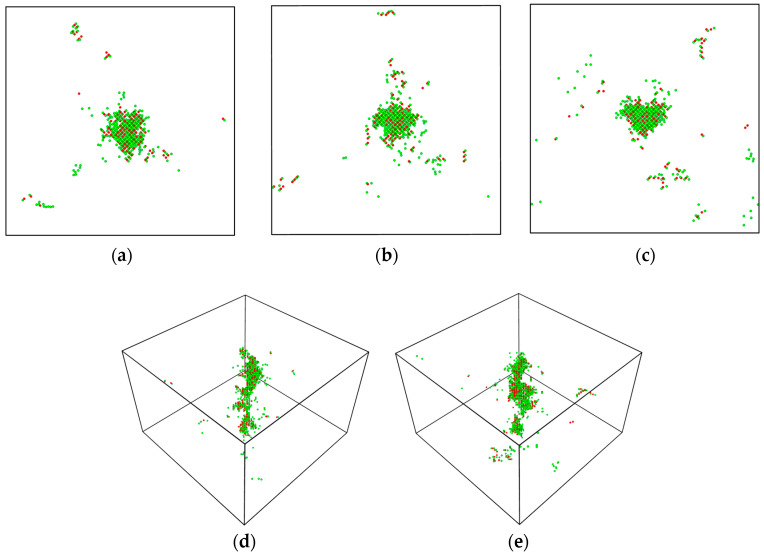
Thermal spike region at different temperatures: (**a**) projection views of 300 K, (**b**) projection views of 400 K, (**c**) projection views of 500 K, (**d**) perspective view of 400 K, and (**e**) perspective view of 500 K; red particles represent magnesium atoms and green particles represent tritium atoms.

**Figure 3 materials-16-03359-f003:**
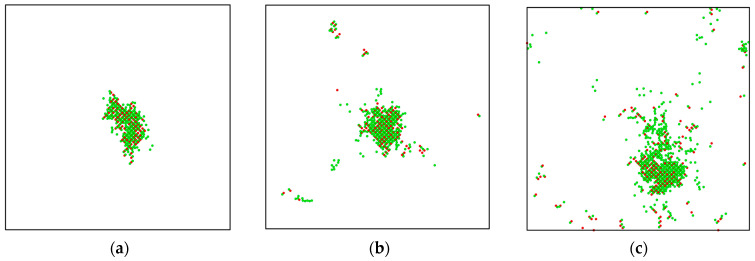
Projection views of thermal spike region at different PKA energies: (**a**) 300 K–1 keV, (**b**) 300 K–2 keV, and (**c**) 300 K–5 keV; red particles represent magnesium atoms and green particles represent tritium atoms.

**Figure 4 materials-16-03359-f004:**
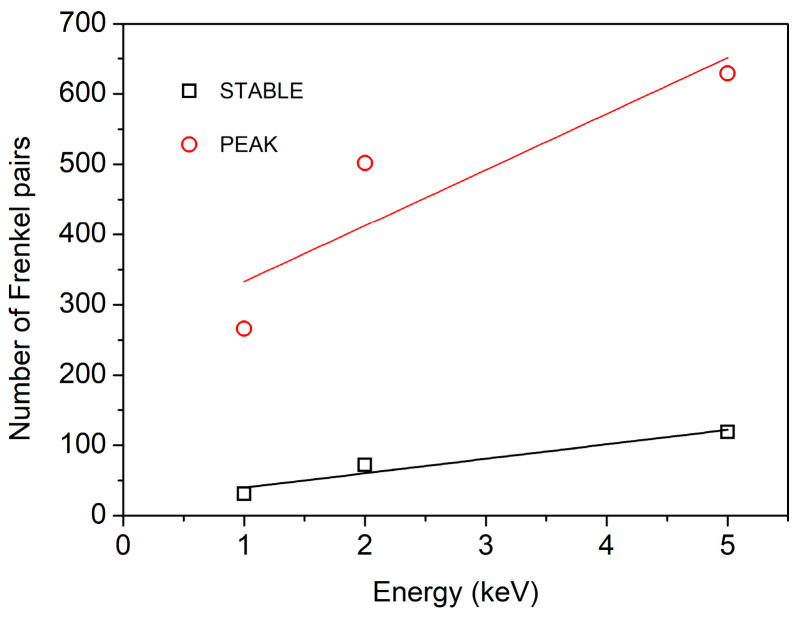
Peak and stable number of Frenkel pairs with different PKA energies, T = 300 K.

**Figure 5 materials-16-03359-f005:**
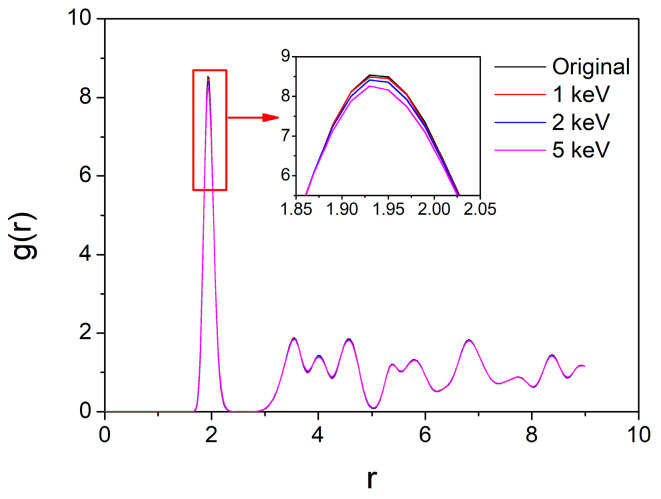
RDF of Mg-T in MgT_2_ before and after radiation damage.

**Figure 6 materials-16-03359-f006:**
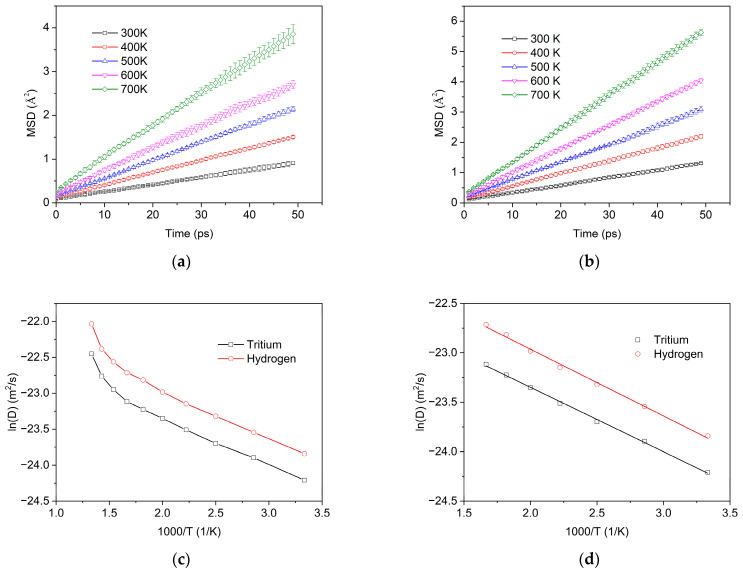
Simulation results of tritium/hydrogen diffusion in perfect MgT_2_/MgH_2_ at different temperatures: (**a**) MSD of tritium, (**b**) MSD of hydrogen, (**c**) Arrenhnius plot of tritium/hydrogen, and (**d**) Arrenhnius fit results of tritium/hydrogen, temperature range from 300 K to 600 K.

**Figure 7 materials-16-03359-f007:**
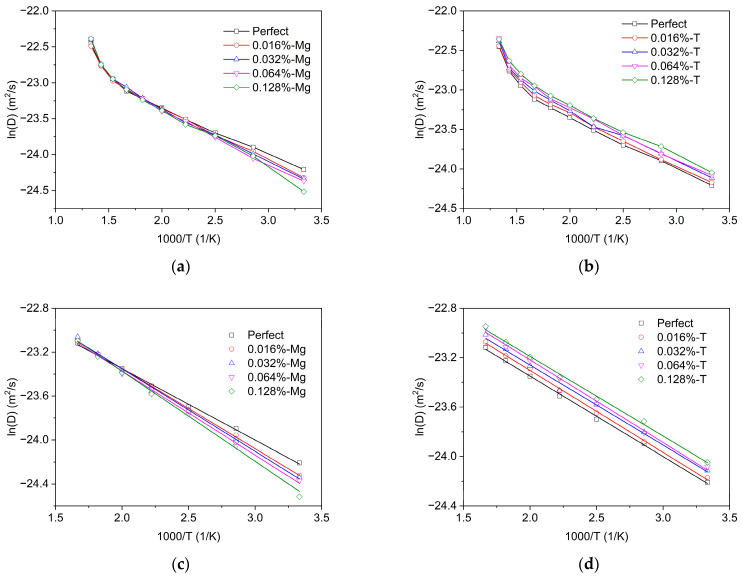
Arrhenius plots of tritium diffusivity in MgT_2_ with different SIA concentrations: (**a**) different concentrations of magnesium (Mg) interstitials, (**b**) different concentrations of tritium (T) interstitials, (**c**) fitting results of magnesium (Mg) interstitials, and (**d**) fitting results of tritium (T) interstitials.

**Figure 8 materials-16-03359-f008:**
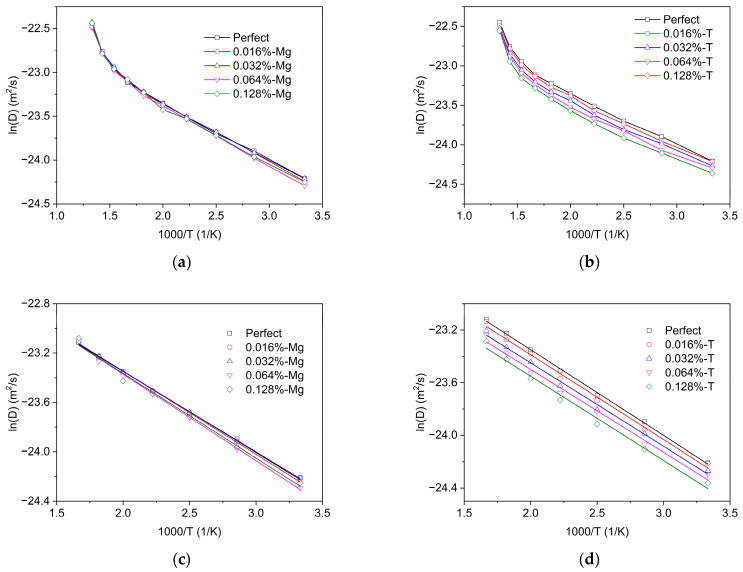
Arrhenius plots of tritium diffusivity in MgT_2_ with different vacancies concentrations: (**a**) different concentrations of magnesium (Mg) vacancies, (**b**) different concentrations of tritium (T) vacancies, (**c**) fitting results of magnesium (Mg) vacancies, and (**d**) fitting results of tritium (T) vacancies.

**Table 1 materials-16-03359-t001:** Diffusivity (*D*) of tritium/hydrogen with temperatures (*T*) ranging from 300 K to 750 K.

*T* (K)	*D* of Tritium (10^−8^ m^2^/s)	*D* of Hydrogen (10^−8^ m^2^/s)
350	0.004187	0.005952
400	0.005100	0.007454
450	0.006157	0.008851
500	0.007227	0.010446
550	0.008197	0.012337
600	0.009129	0.013659
650	0.010815	0.015887
700	0.013026	0.019022
750	0.017751	0.026955

**Table 2 materials-16-03359-t002:** Diffusion prefactors (*D*_0_) and energy barriers (*E_a_*) for the diffusion of tritium were measured for different concentrations of interstitial (*C_SIA_*) over a temperature range of 300 K to 600 K.

*C_SIA_*	*D* _0_	*E_a_*	*C_SIA_*	*D* _0_	*E_a_*
0.016%-Mg	3.12 × 10^−10^	0.063	0.016%-T	2.86 × 10^−10^	0.057
0.032%-Mg	3.25 × 10^−10^	0.065	0.032%-T	2.89 × 10^−10^	0.056
0.064%-Mg	3.27 × 10^−10^	0.066	0.064%-T	3.14 × 10^−10^	0.058
0.128%-Mg	3.65 × 10^−10^	0.071	0.128%-T	3.08 × 10^−10^	0.056

**Table 3 materials-16-03359-t003:** Diffusion prefactors (*D*_0_) and energy barriers (*E_a_*) for the diffusion of tritium were measured for different concentrations of vacancies (*C_v_*) over a temperature range of 300 K to 650 K.

*C_v_*	*D* _0_	*E_a_*	*C_v_*	*D* _0_	*E_a_*
0.016%-Mg	2.79 × 10^−10^	0.058	0.016%-T	2.53 × 10^−10^	0.056
0.032%-Mg	2.72 × 10^−10^	0.057	0.032%-T	2.32 × 10^−10^	0.055
0.064%-Mg	2.87 × 10^−10^	0.060	0.064%-T	2.19 × 10^−10^	0.054
0.128%-Mg	2.81 × 10^−10^	0.059	0.128%-T	2.13 × 10^−10^	0.055

## Data Availability

The raw/processed data required to reproduce these findings can be shared upon request.

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
