# Peer review of "Investigating the Impact of Displacement Cascades on Tritium Diffusion in MgT2: A Molecular Dynamics Study"

_materials, 2023, doi:10.3390/ma16093359_

Round 1
Reviewer 1 Report
Dear authors, thank you for an interesting article. Your research will be of interest to many, as it describes
the differences in the diffusion properties of hydrogen and tritium. However, while reading, I had a few
questions:
1. Tell us in more detail what is the reason for the linear behavior of ln(D) with respect to 1/T.
2. Which section in Figure 1 corresponds to which process? Where is the thermal spike phase, annealing
phase? What is the reason for such formation of the number of Frenkel pairs over time?
3. Please describe Figure 1 (a) in more detail. What could be the reason that the number of Frenkel pairs
have the highest peak values at a temperature of 500 degrees, average at 300 degrees, and the lowest at
400? I think that the peak values should increase with increasing temperature, as it happens in Figure 1(b)
with increasing energy. Describe this moment in more detail.
4. How will the projection views of the thermal spike region look like for samples with different
temperatures: 500K, 400K, 300K?
5. What do the lines shown in Figure 3 mean? Do the authors mean by this that the dependence is linear?
6. Also, what does the box in Chart 4 mean? Improve the quality and increase the font of the axial values
as the insert is illegible.
In addition, I believe that the authors should correct a few shortcomings regarding the design, namely:
1. The quality of all figures needs to be improved.
2. The quality of the second figure should be improved. Make the colored dots brighter and more distinct.
I hope that the above shortcomings will help the authors in completing the work.
Author Response
Dear Editor and reviewers,
We appreciate you for your precious time in reviewing our paper and providing valuable comments. It was your valuable and insightful comments that led to possible improvements in the current version. The authors have carefully considered the comments and tried our best to address every one of them. We hope the manuscript after careful revisions meet your high standards. The authors welcome further constructive comments if any.
Below we provide the point-by-point responses. All modifications in the manuscript have been highlighted in red.
Please see the attachment.
Sincerely,
Dan Xiao, xiaod@inest.cas.cn
Hefei Institutes of Physical Science, Chinese Academy of Sciences

Reviewer 2 Report
In the research article titled “Investigating the Impact of Displacement Cascades on Hydrogen Diffusion in MgT2: A Molecular Dynamics Study” by Zhang et al., authors have presented the good analysis, but there are few issues which I think should be addressed. The highlighted issues are as follow;
1. Authors have performed many measurements but in the abstract portion there is just the general discussion, aurthors should highlight (values) of their achievements in the abstract section.
2. As temperature increases, the number of defects tends to increase; however, the number of stable defects does not exhibit a significant difference. What change occurs when temperature decreased?
3. To evaluate the effect of defects on tritium release, here analyzed the variation of tritium density as a function of distance from magnesium atoms using radial distribution function (RDF), explain how radial distribution function works?
4. How the presence of vacancies affects the diffusion energy barrier? Comment.
Author Response

(The authors gave the same response as above.)

Reviewer 3 Report
The manuscript studies the effect of radiation damage on the alpha-MgT2 material. The effects of temperature and energy of primary knock off atom are discussed. The radiation damage effects are studied in terms of number of induced Frenkel defects and diffusivity of hydrogen or tritium atoms in the matrix. This issue is quite interesting due to practical applications of MgH2 or MgT2 in neutron sources.
There are some minor issues that need to be clarified/improved before the manuscript could be considered for publication:
1. Figure 1b suggests that for higher PKA energy the system is not able to sufficiently relax the induced defects, I suggest that the Authors should discuss whether increase in the simulation time is required to estimate the number of Frenkel defects in stabilized state. Especially in context of discussion of Figure 3.
2. How was the number of self interstitial atoms controlled during the simulations discussed in section 3.2.2? How stable it was during the simulation time? This issue should be discussed in Materials and Methods section. Do the Authors expect that the diffusion of hydrogen or tritium would be affected if both tritium and magnesium position defects were present in the simulated system in the same time?
3. "radius distribution function" should be replaced with "radial distribution function" (page 2 line 85).
4. The statement "The slope of the curve representing the peak defect number is 1.0 times steeper than the slope of the stable number of defect." (page 4 lines 156 -158) is incorrect and should be amended.
5. Statement "hydrogen (the isotope of tritium)" (line 183-184) is incorrect, the tritium is the isotope of hydrogen. It should be corrected.
Author Response

(The authors gave the same response as above.)

Round 2
Reviewer 1 Report
It may be accepted for publication.
Author Response
Dear Reviewer,
We appreciate you for your precious time in reviewing our paper.
Sincerely,
Dan Xiao
